# PRISM: Probabilistic Real-Time Inference in Spatial World Models

**Atanas Mirchev**[1], **Baris Kayalibay**[1], **Ahmed Agha**[1],
**Patrick van der Smagt**[1], **Daniel Cremers**[2], **Justin Bayer**[1]

[1]Machine Learning Research Lab, Volkswagen Group, [2]Technical University of Munich
`atanas.mirchev@argmax.ai`

**Abstract:** We introduce PRISM, a method for real-time filtering in a probabilistic generative model of agent motion and visual perception. Previous approaches either lack uncertainty estimates for the map and agent state, do not run in real-time, do not have a dense scene representation or do not model agent dynamics. Our solution reconciles all of these aspects. We start from a predefined state-space model which combines differentiable rendering and 6-DoF dynamics. Probabilistic inference in this model amounts to simultaneous localisation and mapping (SLAM) and is intractable. We use a series of approximations to Bayesian inference to arrive at probabilistic map and state estimates. We take advantage of well-established methods and closed-form updates, preserving accuracy and enabling real-time capability. The proposed solution runs at 10Hz real-time and is similarly accurate to state-of-the-art SLAM in small to medium-sized indoor environments, with high-speed UAV and handheld camera agents (Blackbird, EuRoC and TUM-RGBD).

**Keywords:** generative model, SLAM, Bayes filter, uncertainty, diff. rendering

## 1 Introduction

Moving agents perceive streams of information, typically a mix of RGB images, depth and inertial measurements. Probabilistic generative models [1] are a principled way to formalise the *synthesis* of this data, and from these models inference can be derived through Bayes' rule. We focus on exactly such inference and target the agent states and the scene map, a problem known as simultaneous localisation and mapping (SLAM). We treat it as a posterior approximation for a given state-space model, such that the combination is useful for model-based control: the posterior inference serves as a state estimator and the predictive state-space model as a simulator with which to plan ahead [2].

To pave the way towards decision making, we believe an inference method should have:

- a compatible predictive model for both RGB-D images and 6-DoF dynamics;
- principled state and map uncertainty;
- real-time performance on commodity hardware;
- state-of-the-art localisation accuracy.

We motivate these requirements further in appendix J. Prominent methods like LSD-SLAM [3], ORB-SLAM [4], DSO [5] have propelled visual SLAM forward, with heavy focus on large-scale localisation. The core of modern large-scale SLAM is maximum a-posteriori (MAP) smoothing in a probabilistic factor graph [6, 7]. At present this demands sparsity assumptions for computational feasibility, which obstructs the tight integration of dense maps and rendering. Nonetheless, for smaller scenes the recent popularity of neural models (e.g. NERF [8]) has sparked interest in inference through a renderer (e.g. [9, 10, 11]), but dynamics modelling and uncertainty have remained out of scope. Conversely, classical filtering comes with dynamics and uncertainty in real-time (e.g. [12, 13, 14]),

6th Conference on Robot Learning (CoRL 2022), Auckland, New Zealand.

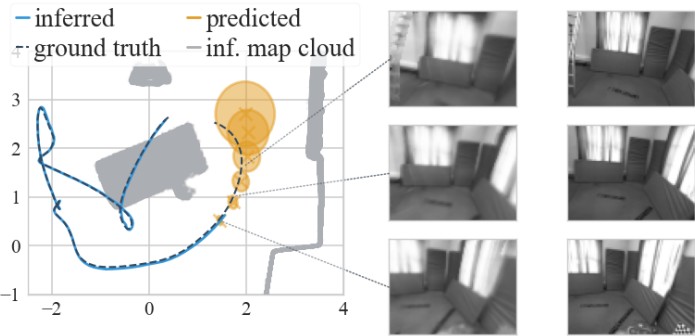

Figure 1: Inference is tailored to the depicted predictive model. Predicting future rollouts, as shown, is required for optimal control. Ground-truth trajectory in *black*, inferred trajectory from past data in *blue*. In *orange*, we see uncertainty envelopes for the predicted future states. On the right, we see predicted and ground-truth future images. Visualised in 2D for clarity, our method operates in 3D.

but over time has given way to large-scale smoothing [6] and to our knowledge has not been well explored for the integration of dense differentiable rendering and dynamics on a moderate scale.

Overall, we find there is a need for a cohesive inference solution that satisfies our requirements. We thus contribute by meeting all the above goals, emphasising the link to a predictive model (fig. 1).

We start from the generative model of Mirchev et al. [15], who combine differentiable rendering and agent dynamics in a probabilistic framework. The authors considered stochastic variational inference for this model, applying it off-line with runtime orders of magnitude too long for on-line use. We pursue an alternative route for real-time inference: from the generative assumptions we derive approximations to the true marginal filters over the last state and map [16]. By focusing on recursive filtering updates, we identify where established probabilistic inference and computer vision techniques can be used, putting emphasis on fast closed-form updates. We find this divide-and-conquer strategy is a good compromise for achieving the aforementioned objectives under computational constraints.

We evaluate the proposed solution on two unmanned aerial vehicle (UAV) data sets [17, 18] and on TUM-RGBD [19]. Our method PRISM runs at 10 Hz real-time with similar localisation accuracy to state-of-the-art SLAM in moderately-sized indoor environments. It provides uncertainty estimates and features a predictive distribution that can both render images and forecast the agent's movement.

## 2 Related Work

**Generative models** Generative state-space models simulate the formation of observed data over time in a Markov chain [1, 12, 20, 21, 22, 23], serving as *world* models [24, 25]. With their agent dynamics and state-to-observation emission models we can imagine future rollouts for planning [2, 26, 27, 28, 29, 30, 31, 32, 33]. We abide by this framework and design a posterior inference for a *spatial* state-space model, to enable on-line control. Among such models (e.g. [34, 35, 36, 37, 38, 39, 40]), we tailor our inference to the model of Mirchev et al. [15]. It scales to 3D with rendering and 6-DoF dynamics. We contribute a real-time inference that fits its probabilistic formulation.

**SLAM through image synthesis** The assumed generative model renders RGB-D images, which is related to SLAM through full-image synthesis. Traditional methods feature varied maps, from volumetric to surfels (e.g. [41, 42, 43, 44, 45, 46, 47, 48, 49]), and commonly estimate new camera poses by aligning new observations to a rendered image with variants of point-to-plane ICP with photometric consistency [50, 51, 52, 53, 44]. We extend this optimisation with dynamics in our approximate state filter [54]. A recent trend is to use implicit scene representations like NERF (e.g. [8, 55, 56]) with high rendering fidelity. Gradient-based pose inference through NERF-like rendering has received attention [57, 58], with iMAP [11] and NICE-SLAM [9] being two real-time

solutions. The mapping runtime of such methods is weighed down by optimisation through the renderer. Rendering can be sped up by decomposing parameters over space, e.g. by using voxels or primitives [59, 60, 61, 62, 63], but how to update neural maps in closed form remains unclear. Therefore, we rely on vanilla voxel grid maps [15, 64], as their probabilistic treatment and closed-form updates are straightforward, leaving implicit representations for future work. We note that none of the aforementioned methods incorporate dynamics and uncertainty, which distinguishes our approach.

**Probabilistic SLAM inference** SLAM filters are thoroughly explored for flat 2D modelling [65, 12, 66, 13, 67, 68, 69], but have been superseded by MAP smoothing in modern visual SLAM (e.g. [3, 4, 5, 70, 71, 72, 73]), primarily due to scalability concerns [6, 74]. However, as of now smoothing is not computationally feasible without sparsity assumptions. We therefore reexamine filtering for differentiable rendering, as we aim to obtain a dense map posterior with uncertainty in real-time (see appendix J for further motivation). Filters may benefit from the dense modelling of observations [74], which aligns with our objective, and we will demonstrate they can be a feasible solution for moderately-sized indoor environments. For the states, we use a Laplace approximation [75] and velocity updates similar to those in extended Kalman filters [12]. For the map, occupancy grids are a common probabilistic choice [64, 66, 76] and closed-form mapping has been used in that context [69]. To enable rendering we provide a similar derivation, but for a signed distance function (SDF), which is related. Probabilistic SDF mapping dates back to Curless and Levoy [41], and SDF updates have a well-known probabilistic interpretation [77, 78]. We use these approximations to arrive at a holistic probabilistic solution that scales to dense 3D modelling in real-time.

# 3 Overview

We approach on-line SLAM inference with two aims in mind. First, we want to harmonise our map and state estimation with a predictive model. Second, we want to quantify uncertainty: estimates and predictions should account for modelling inaccuracies as well as measurement and process noise. Both are important for autonomous decision making. To achieve this, we derive a Bayesian posterior in the probabilistic model of Mirchev et al. [15], to ensure that inference matches the forward model. Before we delve into our proposed solution, we present a practical summary. At every time step:

1. we point-estimate the agent's pose using gradient descent, involving geometry and dynamics.
2. we extend the pose with a Gaussian covariance matrix through a Laplace approximation.
3. with the pose, we estimate the agent's current velocity in closed form.
4. with the pose and the current observation, we update the map in closed form.

We use well-established methods for the above. In 1. we combine assumed density filtering [79], point-to-plane ICP [50] and photometric alignment [51, 52]. In 2. we use a Laplace approximation [75, 80]. In 3. we use linear-Gaussian updates, akin to Kalman filters [12]. In 4. we first derive generic closed-form map updates, which boil down to SDF updates [41] for our generative assumptions.

We contribute by deriving a holistic Bayesian inference from the generative model we started with. In doing so, we identify where traditional techniques are applicable to make a practical algorithm.

# 4 Methods

In the following we will denote generative distributions, true posterior distributions and conditionals with $p(\cdot)$. Respectively, approximate distributions will be denoted with $q(\cdot)$. Approximation steps will be indicated by $\approx$ in equations. We use $q^\phi(\cdot)$ to subsume estimated distribution parameters into $\phi$. A subscript $\cdot_t$ indicates that a variable or a distribution is different at every time step.

## 4.1 Background

We start with an overview of the generative model of Mirchev et al. [15] from which we will derive the inference. We assume a sequence of RGB-D observations $\mathbf{x}_{1:T}$ and a sequence of agent states

$\mathbf{z}_{1:T}$ driven by controls $\mathbf{u}_{1:T-1}$ form a Markovian state-space model. Each observation is constructed from a respective state with a rendering emission model $p(\mathbf{x}_t \mid \mathcal{M}, \mathbf{z}_t)$, where $\mathcal{M}$ is a global latent random variable for a dense map. A transition model $p(\mathbf{z}_t \mid \mathbf{z}_{t-1}, \mathbf{u}_{t-1})$ accounts for the agent dynamics, where $\mathbf{u}_t$ are known acceleration controls. Assuming $\mathbf{z}_1$ is given, the joint distribution is:

$$p(\mathcal{M}, \mathbf{z}_{2:T}, \mathbf{x}_{1:T} \mid \mathbf{u}_{1:T-1}, \mathbf{z}_1) = p(\mathcal{M})p(\mathbf{x}_1 \mid \mathcal{M}, \mathbf{z}_1) \prod_{t=2}^{T} p(\mathbf{z}_t \mid \mathbf{z}_{t-1}, \mathbf{u}_{t-1})p(\mathbf{x}_t \mid \mathbf{z}_t, \mathcal{M}).$$

The map is a 3D voxel grid of occupancy and color–each cell contains four values. The emission is fully-differentiable and performs volumetric raymarching, searching for a unique hit position at a surface along each ray [81]. The transition performs Euler integration, using the acceleration controls and maintained velocity from the latent state. Appendix B and the original paper have the details.

## 4.2 Posterior Choice

First we need to choose which posterior to approximate. For example, Mirchev et al. [15] approximate the full posterior over the map and *all* states $p(\mathcal{M}, \mathbf{z}_{2:T} \mid \mathbf{x}_{1:T}, \mathbf{u}_{1:T-1}, \mathbf{z}_1)$ with variational inference [82]. While generic, this approach is slowed down by rendering at every optimisation step [54], and the inevitable stochastic optimisation demands multiple steps until convergence. In addition, estimating the posterior over all states scatters the optimisation budget across the whole trajectory.

To enable real-time inference we target an alternative posterior, the filter $p(\mathcal{M}, \mathbf{z}_t \mid \mathbf{x}_{1:t}, \mathbf{u}_{1:t-1}, \mathbf{z}_1)$, as the last state belief is enough for planning ahead [2]. Since filters can be updated recursively [16, 80], we can use closed-form updates for fast inference. Still, maintaining the joint distribution is too costly because of the large dense 3D map $\mathcal{M}$.[1] Instead, we approximate the two marginal filters:

$$q_t^\phi(\mathcal{M}) \approx p(\mathcal{M} \mid H_t) = p(\mathcal{M} \mid \mathbf{x}_{1:t}, \mathbf{u}_{1:t-1}, \mathbf{z}_1)$$

$$q_t^\phi(\mathbf{z}_t) \approx p(\mathbf{z}_t \mid H_t) = p(\mathbf{z}_t \mid \mathbf{x}_{1:t}, \mathbf{u}_{1:t-1}, \mathbf{z}_1),$$

where $H_t = \mathbf{x}_{1:t}, \mathbf{u}_{1:t-1}, \mathbf{z}_1$. More details about this modelling choice can be found in appendix A. We draw attention to the shorthand notation $p(\ \cdot\ \mid H_t)$, which will appear again in the following.

## 4.3 Approximate Filtering

For both marginal filters, we will arrive at adequate approximations by reusing the following equation:

$$p(\mathcal{M}, \mathbf{z}_t \mid H_t) \propto p(\mathbf{x}_t \mid \mathbf{z}_t, \mathcal{M}) \int p(\mathbf{z}_t \mid \mathbf{z}_{t-1}, \mathbf{u}_{t-1})p(\mathcal{M}, \mathbf{z}_{t-1} \mid H_{t-1})d\mathbf{z}_{t-1}, \qquad (1)$$

This is a classic recursive expression of the Bayes filter [16]. Starting from each true marginal posterior, we will first expand the joint, then use eq. (1) and apply a set of approximations. Next we will discuss our final result, we defer the detailed derivation of both filters to appendices C and E.

### 4.3.1 Marginal Map Filter

We begin with the map approximation, starting from the true marginal Bayes filter:

$$p(\mathcal{M} \mid H_t) = \int p(\mathcal{M}, \mathbf{z}_t \mid H_t) \, \mathrm{d}\mathbf{z}_t$$

$$\propto \int p(\mathbf{x}_t \mid \mathbf{z}_t, \mathcal{M}) \int p(\mathbf{z}_t \mid \mathbf{z}_{t-1}, \mathbf{u}_{t-1})p(\mathcal{M}, \mathbf{z}_{t-1} \mid H_{t-1}) \, \mathrm{d}\mathbf{z}_{t-1} \, \mathrm{d}\mathbf{z}_t$$

$$\approx p(\mathbf{x}_t \mid \hat{\mathbf{z}}_t, \mathcal{M}) \times q_{t-1}^\phi(\mathcal{M}) \qquad (2)$$

$$\approx q(\mathcal{M} \mid \mathbf{x}_t, \hat{\mathbf{z}}_t) \times q_{t-1}^\phi(\mathcal{M}) =: q_t^\phi(\mathcal{M}). \qquad (3)$$

Equations (2) and (3) hide a few approximations detailed in appendix C. The resulting solution takes a nominal state sample $\hat{\mathbf{z}}_t$, with which a map update $q(\mathcal{M} \mid \mathbf{x}_t, \hat{\mathbf{z}}_t)$ is applied to the previous map belief $q_{t-1}^\phi(\mathcal{M})$. We set $\hat{\mathbf{z}}_t$ to the mean of the current state belief $q_t^\phi(\mathbf{z}_t)$. Accepting some bias, we do this for speed as it is our best guess for $\mathbf{z}_t$ without extra computation.[2] Intuitively, the map update

---

[1]E.g. the size of full-covariance Gaussian representations [12] or carrying multiple maps in parallel for a Rao-Blackwellised particle filter [13, 14] become prohibitive.

[2]Appendix F discusses this approximation further.

$q(\mathcal{M} \mid \mathbf{x}_t, \hat{\mathbf{z}}_t)$ populates the map such that the observation $\mathbf{x}_t$ can be reconstructed. Our derivation of the updates is similar to the one by Grisetti et al. [69] for 2D occupancy maps, but now applied to 3D.

The above approximation is generic, agnostic to the specific map and rendering assumptions. In practice, we need a closed-form map update $q(\mathcal{M} \mid \mathbf{x}_t, \hat{\mathbf{z}}_t)$ that is faithful to the emission $p(\mathbf{x}_t \mid \hat{\mathbf{z}}_t, \mathcal{M})$. In this work, we follow Mirchev et al. [15] and use a Gaussian map that factorises over voxels:

$$q_t^{\phi}(\mathcal{M}) = \prod_{ijk} \mathcal{N}\big(\mathcal{M}_{ijk} \mid \boldsymbol{\mu}_{ijk,t}^{\mathcal{M}}, \mathrm{diag}((\boldsymbol{\sigma}_{ijk,t}^{\mathcal{M}})^2)\big).$$

Here the indices $ijk$ run over voxels in a 3D grid. For this specific representation and the assumed surface-based rendering, we identify that the map update $q(\mathcal{M} \mid \mathbf{x}_t, \hat{\mathbf{z}}_t)$ can be implemented as a probabilistic signed distance function (SDF) update [41]. We provide the technical details in appendix D. SDF updates for voxel maps are a traditional concept in computer vision, and prior work has considered their probabilistic interpretation before [77, 78]. We contribute by identifying the place of such updates in a probabilistic filter that follows the generative model of [15]. A detailed discussion of how the above relates to classical SDF update equations can be found in appendix D.

The above approximations are motivated by the real-time constraint. For example, one could optimise eq. (2) directly with gradient descent through the renderer, but evaluating the emission is expensive and hinders accurate convergence on a budget. This is particularly true when uncertainty estimates are desirable, as optimisation would then be stochastic and gradients noisy [83]. In contrast, the derived one-shot map updates are meant to have a cost similar to emitting just once, while capturing uncertainty as well. We show some of the differences between the two approaches in section 5.3.

### 4.3.2 Marginal State Filter

Similarly, for the state filter we start from the true marginal and arrive at approximations via eq. (1):

$$p(\mathbf{z}_t \mid H_t) = \int p(\mathcal{M}, \mathbf{z}_t \mid H_t) \, \mathrm{d}\mathcal{M}$$

$$\propto \int p(\mathbf{x}_t \mid \mathbf{z}_t, \mathcal{M}) \int p(\mathbf{z}_t \mid \mathbf{z}_{t-1}, \mathbf{u}_{t-1}) p(\mathcal{M}, \mathbf{z}_{t-1} \mid H_{t-1}) \, \mathrm{d}\mathbf{z}_{t-1} \, \mathrm{d}\mathcal{M}$$

$$\approx p\big(\mathbf{x}_t \mid \mathbf{z}_t^{\mathrm{pose}}, \hat{\mathcal{M}}\big) q_t(\mathbf{z}_t^{\mathrm{pose}} \mid \mathbf{u}_{t-1}, H_{t-1}) q_t(\mathbf{z}_t^{\mathrm{vel}} \mid \mathbf{z}_t^{\mathrm{pose}}, \mathbf{u}_{t-1}, H_{t-1}) \qquad (4)$$

$$\approx q_t^{\phi}(\mathbf{z}_t^{\mathrm{pose}}) \times q_t(\mathbf{z}_t^{\mathrm{vel}} \mid \mathbf{z}_t^{\mathrm{pose}}, \mathbf{u}_{t-1}, H_{t-1}) =: \ q_t^{\phi}(\mathbf{z}_t). \qquad (5)$$

We detail all the approximations that lead to eq. (4) in appendix E. In eq. (4) we have three terms: an image reconstruction likelihood, a Gaussian pose prior and a linear Gaussian velocity conditional given a pose. The latter two we obtain analytically with a linear approximation of the transition model and the previous Gaussian belief $q_{t-1}^{\phi}(\mathbf{z}_{t-1})$ (c.f. appendix E). First, using the first two terms of eq. (4) we define a maximum a-posteriori (MAP) objective for pose optimisation:

$$\arg\max_{\mathbf{z}_t^{\mathrm{pose}}} \ \log p\big(\mathbf{x}_t \mid \hat{\mathcal{M}}, \mathbf{z}_t^{\mathrm{pose}}\big) + \log q_t(\mathbf{z}_t^{\mathrm{pose}} \mid \mathbf{u}_{t-1}, H_{t-1}).$$

Here, $\hat{\mathcal{M}}$ is a nominal map sample set to the mean of the previous map belief $q_{t-1}^{\phi}(\mathcal{M})$.[3] The term $\log q_t(\mathbf{z}_t^{\mathrm{pose}} \mid \mathbf{u}_{t-1}, H_{t-1})$ is an approximate dynamics prior over the current pose, it makes the pose respect the transition model. The term $\log p(\mathbf{x}_t \mid \hat{\mathcal{M}}, \mathbf{z}_t^{\mathrm{pose}})$ represents reconstructing the current observation, optimising it for the current pose will align the observation to the map. However, evaluating this rendering term in every gradient step is inefficient. Because of this, we replace it with the prediction-to-observation objective used by Kayalibay et al. [54], Nießner et al. [45], Newcombe et al. [84]. We refer to [54] for further motivation and we list the technical details in appendix E.

The above optimisation gives us a MAP pose estimate, which we denote with $\boldsymbol{\mu}_t^{\mathrm{pose}}$. Next, we apply a Laplace approximation [75] around it to obtain a full covariance matrix $\boldsymbol{\Sigma}_t^{\mathrm{pose}}$ which captures the curvature of the objective. This leaves us with a full Gaussian belief over the current pose:

$$q_t^{\phi}(\mathbf{z}_t^{\mathrm{pose}}) = \mathcal{N}(\mathbf{z}_t^{\mathrm{pose}} \mid \boldsymbol{\mu}_t^{\mathrm{pose}}, \boldsymbol{\Sigma}_t^{\mathrm{pose}}).$$

---

[3]Appendix F discusses this approximation further.

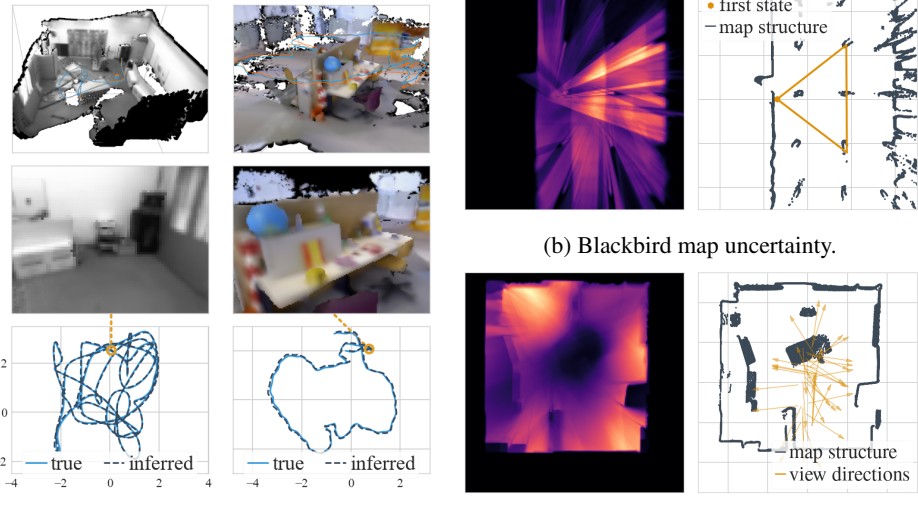

(a) Example mapping and localisation

(b) Blackbird map uncertainty.

(c) EuRoC map uncertainty.

Figure 2: (a) 3D reconstruction, example emission and inferred trajectory for EuRoC/V102 and TUM-RGBD fr3/office. (b) Blackbird experiment. Top-down map uncertainty on the left, *black* is uncertain, *orange* is precise. Precision is highest in a triangle around the center, which is the camera frustum where the agent remains sitting on a platform for a long time, see the orange triangle amidst the map point cloud on the right. (c) Analogous EuRoC experiment. Map uncertainty is high outside of the room, at the center and behind the two structures on the left due to occlusion. The uncertainty in the center is high because the agent primarily looks outwards (view directions in the right image).

Finally, we can combine this Gaussian with the Gaussian velocity conditional $q_t(\mathbf{z}_t^{\text{vel}} \mid \mathbf{z}_t^{\text{pose}}, \mathbf{u}_{t-1}, H_{t-1})$ (the third term in eq. (4)) into a full-state belief in closed form:

$$q_t^{\phi}(\mathbf{z}_t) = \mathcal{N}(\mathbf{z}_t \mid \boldsymbol{\mu}_t, \boldsymbol{\Sigma}_t) = \mathcal{N}(\mathbf{z}_t^{\text{pose}} \mid \boldsymbol{\mu}_t^{\text{pose}}, \boldsymbol{\Sigma}_t^{\text{pose}}) \mathcal{N}(\mathbf{z}_t^{\text{vel}} \mid \mathbf{D}_t \mathbf{z}_t^{\text{pose}} + \mathbf{e}_t, \boldsymbol{\Sigma}_t^{\text{vel}}).$$

This is approximate, we do it for speed and find it does not harm localisation in practice. Appendix E describes how the linear Gaussian terms come to be in more detail.

## 5 Experiments

Originally we set out with a few goals: the inference method should be faithful to the generative assumptions, it should quantify uncertainty and it should run in real-time. What follows is an empirical analysis of these aspects. We evaluate on the EuRoC [17], Blackbird [18] and TUM-RGBD [19] data sets. The agent in the former two is an unmanned aerial vehicle (UAV), with speed of up to 4 m/s. For Blackbird, we use Semi-Global Block Matching (SGBM) for stereo depth estimation [85]. For EuRoC, we use the ground-truth Leica MS50 depth readings provided by [10]. We pretend the IMU readings from these data sets are our control inputs. For TUM-RGBD we do not feed in any controls and assume a constant-velocity transition. All experimental details are in appendix G.

### 5.1 Inference Through a Probabilistic Generative Model

First we look into the synergy between the inference and the generative assumptions. In fig. 2a we see mapping and localisation examples. The inferred scenes are consistent, with no dramatic offsets in geometry. More importantly, rendering from the inferred map using the emission $p(\mathbf{x}_t \mid \mathbf{z}_t, \mathcal{M})$ works as expected (see middle row), indicating that map updates are consistent with the generative assumptions. This is evident from the accuracy of the inferred state trajectories as well (last row), as the pose optimisation objective from section 4.3.2 uses rendered images at every filtering step. A potential discrepancy between the inference and the generative assumptions would lead to errors that would accumulate over time, which is not the case.

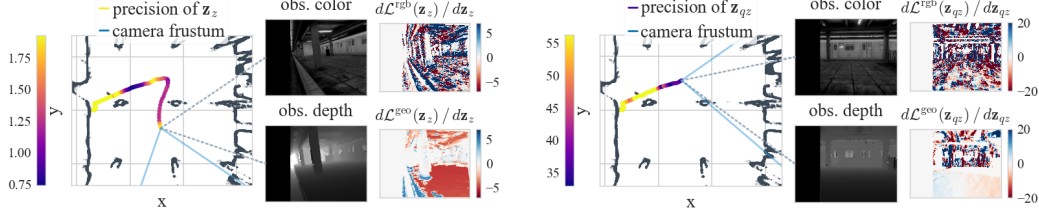

(a) $x$ location uncertainty            (b) $\mathbf{q}_z$ (yaw) orientation uncertainty

Figure 3: Inferred state uncertainty. Inferred trajectories are colored by precision (inverse uncertainty) of a certain state dimension, followed by observations, followed by columns of the tracking Jacobian for that same state dimension. (a) Here the precision in $z$ (vertical movement) is high (*yellow*), because the $z$-orthogonal floor produces a consistent Jacobian (bottom right). (b) Here the precision in $\mathbf{q}_z$ orientation (yaw, azimuth) is low (*violet*), as there are no orthogonal surfaces (i.e. facing sideways). Note the low Jacobian magnitude of the horizontal floor this time (bottom right).

**Map uncertainty**     The inferred map uncertainty is determined by the map updates. We show its interpretable effects in figs. 2b and 2c for two examples, one from Blackbird and another from EuRoC. Our map updates are akin to traditional SDF updates and the main factor that decides whether a map region is certain is how often it was observed. Regions that were occluded by objects, are behind walls or were rarely in view remain uncertain, e.g. as seen in fig. 2c. In contrast, if the agent spends a lot of time looking at a certain map region, the uncertainty there decreases, as seen in fig. 2b.

**State uncertainty**     In fig. 3 we analyse state uncertainty by looking at the variance for individual dimensions. We notice that state uncertainty changes along the trajectory. Uncertainty is determined by what the agent currently sees, based on the geometric relationship between the agent movement and the observed scene (e.g. fig. 3a and fig. 3b). This effect can be explained if we examine the Laplace approximation used to estimate pose covariances. At any given time step, we set the covariance to $\mathbf{\Sigma}_t^{\text{pose}} \approx -\mathbf{H}^{-1} \approx -\left(2\mathbf{J}^T\mathbf{J}\right)^{-1}$. Here $\mathbf{H}$ is the Hessian of the tracking objective at the mean pose estimate and $\mathbf{J}$ is the Jacobian. The Jacobian connects the pose to all image pixel errors. The more consistent Jacobian entries are for a given pose dimension, the smaller the variance for that dimension will be. We refer to appendix H for more details about the map and state uncertainty quality.

### 5.2    Localisation Accuracy

We compare PRISM's localisation to state-of-the-art methods in moderately-sized indoor environments. We consider both baselines with dense maps (TANDEM [10], VSSM-LM [15], iMAP [11], NICE-SLAM [9], CodeVIO [86]) and sparse methods without rendering (ORB-SLAM2 [4], VINS [71], VIMO [70]). The results are in table 1. For the considered trajectories accuracy is comparable to the baselines, with differences of a few centimeters. At the same time, our inference boasts a predictive state-space model with both rendering and dynamics as well as uncertainty estimates, which is not common in the dense visual SLAM literature. Finally, in fig. 4 we see example inferred agent velocities, noting the uncertainty bands. This is possible because we model the agent dynamics.

Our localisation accuracy on Blackbird is better than the off-line variational inference results of VSSM-LM presented by Mirchev et al. [15], and at the same time our solution runs in real-time and also captures uncertainty. This shows the advantages of the proposed divide-and-conquer filtering.

### 5.3    Approximations for Runtime Improvement

All of our approximations are motivated by the real-time constraint, dictating the need for closed-form map updates, a Laplace approximation, linearisation assumptions and a surrogate pose optimisation objective. Figure 5 shows a runtime breakdown for different image resolutions, measured on an

---

[4]Last 10 s are skipped, as the drone hits the ground during landing.

Table 1: Localisation absolute error RMSE in meters on EuRoC [17], Blackbird [18] and TUM-RGBD [19].

| Trajectory | Ours | Code VIO | TANDEM | ORB SLAM2 |
|---|---|---|---|---|
| EuRoC/V101 | 0.041 ($\pm$ 0.002) | 0.05 | 0.09 | **0.031** |
| EuRoC/V102 | 0.035 ($\pm$ 0.002) | 0.07 | 0.17 | **0.02** |
| EuRoC/V103 | **0.042 ($\pm$ 0.002)** | 0.07 | - | 0.048 |
| EuRoC/V201 | **0.037 ($\pm$ 0.001)** | 0.10 | 0.09 | **0.037** |
| EuRoC/V202 | **0.035 ($\pm$ 0.003)** | 0.06 | 0.12 | 0.035 |
| EuRoC/V203 | x | **0.275** | - | x |

| Trajectory | Ours | VSSM LM | VIMO | VINS |
|---|---|---|---|---|
| picasso, 1 m/s | 0.064 ($\pm$ 0.003) | 0.139 | **0.055** | 0.097 |
| picasso, 2 m/s | 0.053 ($\pm$ 0.003) | 0.136 | **0.040** | 0.043 |
| picasso, 3 m/s | 0.061 ($\pm$ 0.003) | 0.120 | **0.043** | 0.045 |
| picasso, 4 m/s | 0.079 ($\pm$ 0.005)[4] | 0.174 | **0.049** | 0.056 |
| star, 1 m/s | 0.089 ($\pm$ 0.007)[4] | 0.137 | **0.088** | 0.102 |
| star, 2 m/s | 0.111 ($\pm$ 0.009) | 0.163 | **0.082** | 0.133 |
| star, 3 m/s | **0.115 ($\pm$ 0.012)** | 0.281 | 0.183 | 0.235 |
| star, 4 m/s | **0.153 ($\pm$ 0.015)[4]** | 0.156 | x | x |

| Trajectory | Ours | iMAP | NICE SLAM | ORB SLAM2* |
|---|---|---|---|---|
| fr1/desk | 0.053 ($\pm$ 0.003) | 0.049 | 0.027 | **0.016** |
| fr2/xyz | 0.029 ($\pm$ 0.001) | 0.02 | **0.018** | 0.04 |
| fr3/office | 0.083 ($\pm$ 0.001) | 0.058 | 0.03 | **0.01** |

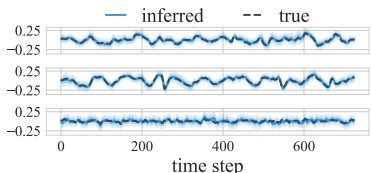

Figure 4: Inferred $xyz$-velocity.

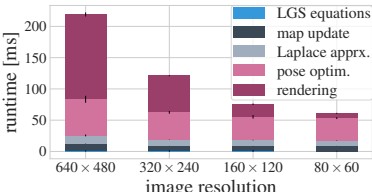

Figure 5: Runtime breakdown.

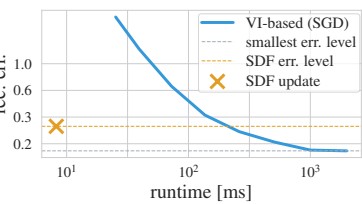

Figure 6: Mapping comparison.

NVIDIA 1080 Ti GPU and an Intel(R) Xeon(R) W-2123 CPU at 3.6 GHz. The heaviest operations are rendering and the gradient-based pose optimisation. Based on movement speed, rendering can happen periodically, whenever a new anchor image prediction for pose optimisation is needed. This leaves us with a total runtime of 10 Hz to 15 Hz, updating the map and state at every data step. In fig. 6 we also compare closed-form map updates to map inference via gradient-descent (e.g. as in [15, 8, 11, 9]). While gradient-descent is more accurate on a bigger budget, it is much more expensive. For example, to match the accuracy of the closed-form updates, which take less than 10 ms, one would need ca. 250 ms of optimisation, which is impractical. These runtimes are for a voxel grid that is significantly faster than neural representations [54], which would only exacerbate the problem.

## 6 Limitations and Conclusion

SDF voxel grids allow for closed-form updates, but their memory footprint limits the maximum resolution and scene size. Voxel hashing [45] or octrees [87] can directly replace them for memory efficiency. Neural maps and dynamically changing maps have remained out of our scope. Their probabilistic formulation and closed-form updates require further investigation. Our map factorises over voxels with no inter-region correlation, which could also be improved. PRISM provides interpretable uncertainty in real-time, but estimation is approximate. Obtaining perfectly calibrated uncertainty on a budget remains an open question (see appendix H). While filtering works for our generative assumptions indoors, filters cannot revisit past errors and can drift in large scenes with high levels of exploration [6]. We leave large-scale inference considerations for future work.

We have introduced PRISM, a method for probabilistic filtering in a predefined spatial state-space model. Our solution runs in real-time, provides state and map uncertainty, and infers a dense map and a 6-DoF state trajectory with velocities. It is comparably accurate to state-of-the-art SLAM in indoor environments. To the best of our knowledge this is the first real-time fully-probabilistic solution for SLAM that combines differentiable rendering and agent dynamics. We validated our method on three challenging data sets, featuring unmanned aerial vehicles and a handheld camera. The results are promising, establishing PRISM as a viable state estimator for downstream model-based control.

**Acknowledgments**

We thank our reviewers for the thoughtful discussion, it helped us to better position our contribution.

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
