# OpenReview forum: "PRISM: Probabilistic Real-Time Inference in Spatial World Models"
_robot-learning.org/CoRL/2022/Conference — CoRL 2022 Oral_

### Official Review · Reviewer_6pXK · 2022-07-20

**Originality:** Good
**Technical Quality:** Very Good
**Clarity Of Presentation:** Good
**Impact:** 3

**Recommendation:**

Weak Accept: I recommend accepting the paper, but will not argue for my recommendation if the majority of other reviewers have a different opinion.

**Summary:**

This paper presents a filtering approach for real-time 3D mapping and localization with a dense geometric map representation. The approach leverages a simple and classical heuristic, which is to decouple the map and robot state updates entirely. This enables fast, closed-form updates. The system is demonstrated on several benchmark datasets and has competitive estimation accuracy with state-of-the-art approaches.

**Issues:**

See Weaknesses for more detailed comments above; main points reproduced below:

- Clarification: The map cells $\mathcal{M}_{ijk}$ have a Gaussian distribution with diagonal covariance, but what is the data in each cell that we are assuming has Gaussian distribution? What is its dimensionality? This should be made precise.
- The language throughout could be made significantly clearer and the language could be simplified. For example, in the abstract: "We start from a predefined world model which combines differentiable rendering and 6-DoF dynamics" - what is meant by "world model" here? Likewise, in the introduction, we have: "Our motivation for obtaining the inference procedure as the inversion of a well-defined forward model is that their combination lends itself to model-based control, where the former serves as a state estimator and the latter as a simulator with which to plan ahead" which I found difficult to parse (what is the "former" and "latter" referring to here?). I would suggest taking a full pass through the paper and asking for each sentence "How can this be made simpler?"
- The marginal updates appear to be discarding significant uncertainty information. Essentially, the current presentation is directly combining "mapping while assuming known poses" and "localization assuming a known map." The consequence of this is that in scenarios where significantly more robot state/map uncertainty has accumulated, measurements from uncertain states will have equal influence to those from certain states on the quality of the map (and vice versa).
- Because of this issue with uncertainties, an interesting demonstration would be to expand the spatial and temporal scale of experiments (e.g. demonstrate on a larger-scale dataset with more "exploration"). Since this approach doesn't have an obvious "global" loop closure component, I could see this being challenging, and it would be interesting to examine at what point the performance of the estimator degrades.
- Not a weakness, but in light of the above considerations, it may be that this method is best suited to persistent operation is a relatively well-constrained environment where uncertainty accumulation can be limited by the spatial extent of navigation. Alternatively, one could imagine (in a less compute-constrained setting) using this method as a sort of low-computational-cost "odometry" method in conjunction with a SLAM system and maintaining correlations between the reference frames for _submaps_ built using the proposed mapping method (e.g. in a pose-graph optimization type approach). This could help scale this approach to larger environments.
- Minor comment: it seems common in learning and inference literature to use $z$ to denote hidden states of a Markov model (i.e. the robot poses in a SLAM context) and $x$ to denote measurements, which is done here. It is more common in robotics literature to use $z$ to denote *measurements* and $x$ to denote the robot poses. There is no ambiguity in the current presentation, as these terms are clearly defined, but adopting the latter notation convention may make this work easier to parse at a glance for robotics researchers.

**Quality Of The Limitations Section:**

Additional details required

**Reviewer Expertise:**

3: The reviewer is fairly confident that the evaluation is correct

**Robotics Focus:**

Sufficient demonstration on hardware

**Strengths And Weaknesses:**

# Strengths

- The split filter idea is an interesting one, and it is worth noting this is actually a *classical* idea in robot perception, with a rich history dating back to early SLAM literature. However, it also has some known issues (see Weaknesses).

- The approach is _fast_, operating in real-time with realistic hardware considerations, while maintaining a dense geometric map representation, which is the primary motivation.

- The experimental evaluation is fairly thorough.

- The empirical results on some benchmark SLAM sequences are good and competitive with state-of-the-art approaches.

# Weaknesses
(Note I am combining here weaknesses with clarifications / issues)

- Clarification: The map cells $\mathcal{M}_{ijk}$ have a Gaussian distribution with diagonal covariance, but what is the data in each cell that we are assuming has Gaussian distribution? What is its dimensionality? This should be made precise.
- The language throughout could be made significantly clearer and the language could be simplified. For example, in the abstract: "We start from a predefined world model which combines differentiable rendering and 6-DoF dynamics" - what is meant by "world model" here? Likewise, in the introduction, we have: "Our motivation for obtaining the inference procedure as the inversion of a well-defined forward model is that their combination lends itself to model-based control, where the former serves as a state estimator and the latter as a simulator with which to plan ahead" which I found difficult to parse (what is the "former" and "latter" referring to here?). I would suggest taking a full pass through the paper and asking for each sentence "How can this be made simpler?"
- The marginal updates appear to be discarding significant uncertainty information. Essentially, the current presentation is directly combining "mapping while assuming known poses" and "localization assuming a known map." The consequence of this is that in scenarios where significantly more robot state/map uncertainty has accumulated, measurements from uncertain states will have equal influence to those from certain states on the quality of the map (and vice versa).
- Because of this issue with uncertainties, an interesting demonstration would be to expand the spatial and temporal scale of experiments (e.g. demonstrate on a larger-scale dataset with more "exploration"). Since this approach doesn't have an obvious "global" loop closure component, I could see this being challenging, and it would be interesting to examine at what point the performance of the estimator degrades.
- Not a weakness, but in light of the above considerations, it may be that this method is best suited to persistent operation is a relatively well-constrained environment where uncertainty accumulation can be limited by the spatial extent of navigation. Alternatively, one could imagine (in a less compute-constrained setting) using this method as a sort of low-computational-cost "odometry" method in conjunction with a SLAM system and maintaining correlations between the reference frames for _submaps_ built using the proposed mapping method (e.g. in a pose-graph optimization type approach). This could help scale this approach to larger environments.
- Minor comment: it seems common in learning and inference literature to use $z$ to denote hidden states of a Markov model (i.e. the robot poses in a SLAM context) and $x$ to denote measurements, which is done here. It is more common in robotics literature to use $z$ to denote *measurements* and $x$ to denote the robot poses. There is no ambiguity in the current presentation, as these terms are clearly defined, but adopting the latter notation convention may make this work easier to parse at a glance for robotics researchers.

## Details regarding uncertainty

In the map update step (eq (2)-(3)), we have the approximation (dropping the $\phi$ superscript and bold font):

$p(\mathcal{M} \mid H_t) \approx p(x_t \mid \hat{z_t}, \mathcal{M}) \times q_{t-1}(\mathcal{M}) \triangleq q_{t}(\mathcal{M})$.

Since $p(x_t \mid \hat{z_t}, \mathcal{M})$ (likewise the inverse model $q(\mathcal{M} \mid x_t, \hat{z_t})$ that is actually used) is based on a point estimate, none of the uncertainty from the belief in the state $z_t$ is reflected in the updated map distribution $q_{t}(\mathcal{M})$. Following this reasoning recursively (until we arrive at the initial map believe $p(\mathcal{M})$, it's easy to see that uncertainty in $z_{1:t}$ is not reflected in the map uncertainty. Crucially, this is _not_ true of the actual marginal filter. Because of this point estimate approach, the update (in the aligned equations leading up to eq (2)) is perhaps not best thought of as approximating the marginal, but rather as _conditioning_ on the state $\hat{z_t}$. This means the mapping procedure effectively assumes "known" poses (where those poses are derived from the state filter). The state filter, however, does the same conditioning trick (conditioning on the current estimate of the map at time $t$); therefore it is likewise discarding uncertainty in the map. The practical motivation for this approach is understandable: the actual marginal filters for the states/map (or the joint filter) would require maintaining correlations between robot states and map elements, and with a dense map representation this is too computationally expensive for real-time operation. However, the practical _consequence_ of this for a SLAM system is that in longer-term mapping and localization scenarios (particularly during exploration, where drift is a significant issue), the quality of the map and localization gets worse and worse *and* it is difficult to correct later after e.g. a loop closure.

It's worth noting that early work on SLAM focused on "decoupling" landmark and robot state uncertainty for the same computational reasons observed here.
See Durrant-Whyte and Bailey's "Simultaneous Localization and Mapping (SLAM) Part 1" for a contemporaneous review. The following quote from their paper is relevant:
"Most importantly, it was recognized that the correlations between landmarks, that most researchers had tried to minimize, were actually the critical part of the problem and that, on the contrary, the more these correlations grew, the better the solution."

**Summary Of Recommendation:**

The paper is good overall and I believe the proposed systems seems interesting and useful. The experimental demonstration is convincing. Some minor aspects of the presentation could be improved (see Issues), and in particular the scenarios under which the approximations are reasonable (and their effect on uncertainty estimates) should be clarified.

---

### Official Review · Reviewer_ZggF · 2022-07-31

**Originality:** Good
**Technical Quality:** Good
**Clarity Of Presentation:** Excellent
**Impact:** 3

**Recommendation:**

Weak Accept: I recommend accepting the paper, but will not argue for my recommendation if the majority of other reviewers have a different opinion.

**Summary:**

This paper describes a real-time filtering-based approach to simultaneous localization and mapping.  Notable features include an environmental model that captures dense geometry and appearance (using a volumetric grid representation), a differentiable rendering-based measurement model, and a 6DOF (constant-velocity) agent dynamics model that permits prediction of the agent’s future states, which is useful for downstream planning and control tasks (such as flight control of UAVs).  The proposed method is evaluated on 3 benchmark datasets involving AUV and handheld cameras (EuRoC, Blackbird, and TUM-RGBD), where it achieves real-time (10 Hz) performance and localization accuracy comparable with current state of the art SLAM solutions.


**Issues:**

* The characterization of prior work on SLAM should be clarified, and the positioning of this work with respect to prior art should be updated accordingly.

* The limitations section should explicitly address the well-known limitations of filtering-based SLAM methods versus full smoothing approaches, and their ramifications for circumstances under which PRISM would provide an acceptable solution (i.e. these limitations would not *necessarily* be a significant issue over small spatial scales, as the experiments in the current draft show).

* To support a claim that PRISM “achieves localization accuracy comparable with state of the art SLAM solutions”, it would be necessary to experimentally evaluate PRISM’s localization performance after closing loops on a much larger spatial scale.  Ideally, the experimental results would be expanded along these lines, but failing that, the paper should discuss the limitations of the current small-scale test cases in measuring this (potential) discrepancy.


**Quality Of The Limitations Section:**

Limitations are addressed clearly

**Reviewer Expertise:**

4: The reviewer is confident but not absolutely certain that the evaluation is correct

**Robotics Focus:**

Sufficient demonstration on hardware

**Strengths And Weaknesses:**

One of the paper’s most prominent strengths is its careful and principled development of the proposed method.  At the beginning, the authors set themselves the task of devising a SLAM method that meets the following conditions:


* Its environmental model includes dense geometry and appearance information
* It provides a 6DOF model of the agent’s dynamics (which is useful for downstream prediction, planning, and control tasks)
* It provides explicit and principled measures of uncertainty for its state and map estimates
* It achieves real-time performance on commodity hardware

Given these desiderata, the paper then proceeds step-by-step, starting from the general (abstract) probabilistic formulation of the SLAM problem, to propose (i) specific representations for the map and state that satisfy the above conditions, (ii) suitable measurement and dynamics models (built on these representations), and (iii) approximate Bayesian updates for the map and state that permit the overall inference to be realized as a real-time filtering-based SLAM system.

I especially appreciated the paper’s careful attention to detail when developing the proposed method.  The appendices in the supplementary material provide detailed derivations of each of the main results presented in the paper, including careful signposting of what approximations are being made where, and the rationale for doing so (with respect to the goals outlined above).  As a result the technical approach is well-founded and exceptionally clearly explained, and the experimental results demonstrate that the method achieves the stated objectives, specifically real-time operation and localization accuracy comparable with state of the art alternatives (at least for the three test cases considered in the experimental section – more on this below).

In terms of weaknesses:

**The paper significantly mischaracterizes prior work on SLAM, including current state of the art alternatives.**  The paper repeatedly stresses that one of its central aims is to devise an inference method that provides principled uncertainty estimates for the state and the map, claiming this as one of its top-line contributions.  In so doing, it repeatedly attempts to draw a contrast between what it calls “classical probabilistic methods” (which it appears – based on the references it associates with this phrase – to identify with either *EKF-* or *particle filtering*-based approaches specifically) and “modern visual SLAM”.  For example, in its discussion of prior work, the paper asserts that “modern visual SLAM has drifted away from the probabilistic paradigm”.

This assertion is emphatically false.  It is true that modern visual SLAM methods are not based on *filtering* techniques, but this is because over the last ~10-15 years it has become well-understood that optimization-based *full smoothing* methods provide superior accuracy (because they do not suffer from particle depletion in high-dimensional estimation problems as particle filters do, nor irreversible linearization errors as EKFs do) and frequently faster computation (because they can take advantage of the sparsity present in the full smoothing optimization problem, sparsity which is *not* present in the mean-covariance state representation used in standard extended Kalman filters [and which this paper identifies as a key computational bottleneck, motivating the use of the independent marginal posterior filters for the state and map proposed in this work]).

From a probabilistic standpoint, these graphical full smoothing methods perform maximum a posteriori (MAP) inference over the  joint distribution for the SLAM problem shown at the bottom of page 3: that is, they emphatically *are* probabilistically-grounded approaches.  Indeed, these approaches are typically developed by positing a probabilistic generative model for the data (just as this paper does), and then deriving from this the corresponding form of the MAP estimator.  Moreover, having calculated a MAP estimate, it is possible (and in fact common practice) to extract estimates of the posterior uncertainty using *exactly the same* Laplace approximation proposed in this work – except in the case of smoothing approaches, this is an approximation to the *full posterior* distribution, and therefore provides access to posterior covariances among *arbitrary subsets* of variables (something that the method proposed in this work actually cannot do, because it maintains separate marginal filters for the map and state).

**Possible additional limitations of the approach and experiments not addressed:**  One limitation of the proposed approach is that its map representation is based on a voxel grid, which limits the scale of the environments that can be mapped at a given resolution (due to memory consumption).  As a result, the datasets that are used for the experimental evaluation cover a fairly small area by the standards of modern SLAM systems (roughly the size of a large single room – for example, the Blackbird dataset is an environment of size 11 x 11 x 5 meters).

This has important ramifications for the claim that PRISM achieves a “localization accuracy comparable to current state of the art SLAM systems.”  It is well-known that filtering approaches (like PRISM) have difficulty correcting for significant drift (in both the map and state estimates) after closing large loops (basically because doing this precisely would entail propagating corrections backwards through previous state estimates that have already been marginalized out of the filter).  In contrast, full smoothing approaches preserve (and can therefore *correct*, via nonlinear optimization) a joint estimate for the map and *all* current and previous states, and so tend to be much better at closing large loops.

Given these considerations, it is not clear to me whether the experiments reported in Section 5 are actually sufficient to reveal significant differences in localization performance that *may in fact exist* between PRISM and current state of the art alternatives – in particular smoothing-based approaches like ORB-SLAM – as these significant differences may only become apparent when closing loops over much larger spatial scales.

Please see the following references for a more in-depth discussion of all of the above points:

* Dellaert, Frank, and Michael Kaess. "Factor graphs for robot perception." Foundations and Trends in Robotics 6.1-2 (2017): 1-139.

* Kaess, Michael, and Frank Dellaert. "Covariance recovery from a square root information matrix for data association." Robotics and autonomous systems 57.12 (2009): 1198-1210.

Finally, while not (strictly speaking) a “weakness”, I think **the paper could be greatly strengthened by including a more pointed discussion of PRISM’s potential *impact***.

As the paper itself makes clear, most of the features present in PRISM are afforded by alternative techniques: graphical smoothing methods are built on probabilistic models, provide posterior uncertainty estimates, and can easily incorporate 6DOF agent dynamics [simply by specifying the appropriate conditional distributions p(x_{t+1} | x_t, u_t)].  Moreover, multiple works already combine graphical smoothing approaches with (joint) estimation of dense geometry and appearance models (see for example the recent work on Kimera, which is also missing as a reference from the current draft).  In my view the differentiable rendering measurement model proposed in this work may actually be its most distinguishing feature, but it’s not immediately clear to me what the practical advantages of this would be versus e.g. the mesh-based models used in Kimera.


I understand that one of the technical goals of this project was to produce a method that combines all of the aforementioned features into a single estimation procedure, but what is *not* obvious to me is: what is the actual *impact* of having done this?  That is: what *specifically* does (or might) PRISM enable me to do that I could not have achieved by some other means?  [For example, I could easily imagine combining ORB-SLAM for visual-inertial state estimation (with agent dynamics) with some other system for dense mapping to obtain a (more loosely-coupled but) seemingly largely equivalent system using off-the-shelf components.  Why should I prefer PRISM to this alternative?  In particular, is there some *specific* application PRISM enables that I would not be able to tackle with this (straw-man) alternative?]


* Rosinol, Antoni, et al. "Kimera: From SLAM to spatial perception with 3D dynamic scene graphs." The International Journal of Robotics Research 40.12-14 (2021): 1510-1546.


**Summary Of Recommendation:**

 While the paper’s technical development is carefully executed and explained exceptionally clearly, it also mischaracterizes the capabilities of prior state of the art SLAM methods, and therefore the positioning of this work with respect to prior art.  It also lacks experimental evidence necessary to validate one of the central claimed contributions (that PRISM’s localization accuracy is comparable to existing state of the art SLAM methods).

---

### Official Review · Reviewer_uCb5 · 2022-07-31

**Originality:** Good
**Technical Quality:** Excellent
**Clarity Of Presentation:** Excellent
**Impact:** 3

**Recommendation:**

Weak Accept: I recommend accepting the paper, but will not argue for my recommendation if the majority of other reviewers have a different opinion.

**Summary:**

This paper proposes computationally tractable methods and approximations to allow for inference of generative 3D spatial models in real-time.

**Issues:**

See above.

**Quality Of The Limitations Section:**

Limitations are addressed clearly

**Reviewer Expertise:**

3: The reviewer is fairly confident that the evaluation is correct

**Robotics Focus:**

Sufficient demonstration on hardware

**Strengths And Weaknesses:**

Strengths:

This paper demonstrates a clear method for tractable inference of generative spatial models.  The design choices and approximations that are made in order to achieve real-time performance are well-motivated and provide interesting discussions on context with previous work.  Overall, the paper was an enjoyable read, and presents an interesting view on the current state of real-time world models.

Weaknesses:

There is no quantitative evaluation of the uncertainties.  For example, there should be a comparison with the offline method [12], which is the most related baseline.  Section 5.2 and Figure 3 make sense intuitively, but do not demonstrate whether these are meaningful.

The parameters are not unified across all maps while for baselines this may be the case.  A fair comparison would be to use a single set of parameters across environments, or at least show this along with the current evaluation.

For pose estimation, the scales of Laplace also vary across datasets.  Why can we trust these units in general? Depth and RGB losses have very different units, and photometric error uncertainties do not necessarily convey meaningful uncertainty on pose depending on units.

Additional comparisons with [12], or an equivalent offline version of the proposed work should be shown.  [12] learns agent dynamics.  Is the filter version still compatible?  Ultimately, is this method useful for learning models beyond just single-session agent pose and map states?

What are the benefits of the generative model vs. traditional SLAM representations? For a dense TSDF map and a motion model, future states could be predicted in a similar manner, and the Laplace approximation could be used to estimate pose uncertainty, while the TSDF already has weights which are essentially map uncertainties.

Other comments:

What is currently limiting map accuracy? It would be good to perform comparisons with higher image resolution and higher map resolution even if no longer real-time so that one could see whether design choices or the approximation gaps are the limiting factor.

Additional description on the implication of marginal filters would be beneficial.  How is this not assuming independence?  What do we lose by not maintaining the joint distribution?  Why is the mean-field approximation worse when it explicitly represents the joint?

Why is pose optimization first-order? Could this not be done using a second-order optimizer (such as Gauss-Newton)?

Why are the Laplace approximations noisy and require smoothing?

Last line of page 20 has a typo, “exponential moving avarge”.

**Summary Of Recommendation:**

Overall, the paper is clear, well-motivated, and demonstrates interesting methods to achieve tractable inference of 3D spatial models.  In particular, the discussions on where approximations are made and how it draws on previous work is a valuable contribution.  As mentioned above, there are some additional comparisons and improvements that should be made to justify the method's claims, such as the uncertainty quality and comparisons to non-filtering inference methods.

---

### Meta-Review · Area_Chair_j1rw · 2022-08-04

**Recommendation:** Accept (Oral)
**Confidence:** 5

**Metareview:**

Summary and Significance: This paper describes a real-time filtering-based approach to simultaneous localization and mapping. Notable features include an environmental model that captures dense geometry and appearance (using a volumetric grid representation), a differentiable rendering-based measurement model, and a 6DOF (constant-velocity) agent dynamics model that permits prediction of the agent’s future states, which is useful for downstream planning and control tasks (such as flight control of UAVs). The proposed method is evaluated on 3 benchmark datasets involving AUV and handheld cameras (EuRoC, Blackbird, and TUM-RGBD), where it achieves real-time (10 Hz) performance and localization accuracy comparable with current state of the art SLAM solutions.


Pros
This study shows how to infer generative spatial models. Design choices and approximations to attain real-time performance are well-motivated and provide fascinating context with past work. The research is an interesting look at real-time world models with  detailed geometry and aesthetics. It gives a 6DOF model of the agent's dynamics valuable for prediction, planning, and control. It provides explicit and principled state and map uncertainty measurements.
The paper proposes specific representations for the map and state that satisfy the above conditions, suitable measurement and dynamics models (built on these representations), and approximate Bayesian updates for the map and state that allow the overall inference to be realized as a real-time filtering-based SLAM system.
 Extensive derivations of each of the paper's main outcomes, detailing when approximations are explained. The technical approach is well-founded and clearly presented, and experimental results show that the system accomplishes the stated objectives, - real-time operation and localization accuracy similar to state-of-the-art  for the three test cases considered.

Cons
For baselines, parameters may be unified across all maps. A fair comparison would employ a single set of parameters across situations, or at least show this with the evaluation this was not demonstrated. As a baseline, the offline technique [12] should be compared.
  Depth and RGB losses have distinct units, thus photometric error uncertainties may not express pose uncertainty based on units, this was not discussed
It is not clear if  Is this strategy useful for learning models beyond agent position and map state.
The study claims that one of its major contributions is an inference approach that yields principled uncertainty estimates for the state and the map. In doing so, it constantly contrasts "traditional probabilistic methods" . Hence the authors should discuss the  benefits of their generative model over traditional SLAM.
The marginal updates seem to ignore uncertainty data.
It is not clear if the paper sees "Mapping while assuming known postures" and "localization assuming a known map" are 2 different entities.
The claim that Modern visual SLAM methods are not based on filtering techniques should be thoroughly researched to substantiate the claim being made.

Quality : Good
Clarity : Well presented and easy to read
Originality : Very good, there is some incremental work presented but the percentage of original work is high.

Update:
We thank the authors for the revised submission and the updates made to the experimental section which does improve the quality of the paper.



**Best Paper Nomination:**

No